



# INFERNO-peat v1.0.0: A representation of northern high latitude peat fires in the JULES-INFERNO global fire model

Katie R. Blackford[1,2], Matthew Kasoar[1,2], Chantelle Burton[3], Eleanor Burke[3], Colin Prentice[1,4], Apostolos Voulgarakis[1,2,5]

[1] The Leverhulme Centre for Wildfires, Environment, and Society, Imperial College London, SW7 2BX, UK

[2] Department of Physics, Imperial College London, SW7 2BX, UK

[3] Met Office Hadley Centre, Exeter, EX1 3PB, UK

[4] Department of Life Sciences, Imperial College London, SW7 2BX, UK

[5] Atmospheric Environment and Climate Change Laboratory, Technical University of Crete, Kounoupidiana, 73100, Greece

*Correspondence to*: Katie R. Blackford (k.blackford19@imperial.ac.uk)

**Abstract.** Peat fires in the Northern high latitudes have the potential to burn vast amounts of carbon rich organic soil, releasing large quantities of long-term stored carbon to the atmosphere. Due to anthropogenic activities and climate change, peat fires are increasing in frequency and intensity across the high latitudes. However, at present they are not explicitly included in most fire models. Here we detail the development of INFERNO-peat, the first parameterisation of peat fires in

the JULES-INFERNO fire model. INFERNO-peat utilises knowledge from lab and field-based studies on peat fire ignition and spread to be able to model peat burnt area, burn depth and carbon emissions, based on data of the moisture content, inorganic content, bulk density, soil temperature and water table depth of peat. INFERNO-peat improves the representation of burnt area in the high latitudes, with peat fires simulating on average an additional 0.305 M km$^2$ of burn area each year, emitting 224.10 Tg of carbon. Compared to GFED5, INFERNO-peat captures ~20% more burnt area, whereas INFERNO

underestimated burning by 50%. Additionally, INFERNO-peat substantially improves the representation of interannual variability in burnt area and subsequent carbon emissions across the high latitudes. The coefficient of variation in carbon emissions is increased from 0.071 in INFERNO to 0.127 in INFERNO-peat, an almost 80% increase. Therefore, explicitly modelling peat fires shows a substantial improvement in the fire modelling capabilities of JULES-INFERNO, highlighting the importance of representing peatland systems in fire models.

## 1 Introduction

Peatlands are a globally important store of carbon, housing approximately one third of the world's soil carbon despite only covering 3% of the Earth's land surface (Xu et al., 2018; Yu et al., 2010). The high latitudes make up the vast majority of global peatland area, with ~50% occurring in Canada and Russia alone (UNEP, 2022). The northern high latitudes are therefore critical carbon stores containing 415 PgC (Hugelius et al., 2020), exerting a net cooling effect on the atmosphere

(Frolking and Roulet, 2007). Peatlands are rich in carbon as peat forms under waterlogged anaerobic conditions, which reduces the decomposition rates of vegetation, allowing for the build up of carbon rich organic matter within the soil.



However, peatlands are being increasingly threatened by both climate change and anthropogenic activity, with 54% of high latitude peatlands drying over the last 200 years (Zhang et al., 2022). Peatlands are anticipated to continue to degrade with climate change, amplifying carbon loss with the potential to switch peatlands from being sinks to sources of carbon, creating

a positive feedback loop (Loisel et al., 2020; Swindles et al., 2019; Hugelius et al., 2020; Tarnocai, 2009; Zhao and Zhuang, 2023). Furthermore, the degradation of peatlands resulting from both humans and climate change is increasing the frequency and extent of wildfires in peatlands (Turetsky et al., 2015; Dadap et al., 2019).

Peat fires are among the largest and most persistent wildfire phenomena on Earth (Rein, 2013). In the northern high latitudes,

peat fires largely originate from lightning strikes (Wendler et al., 2011; McCarty et al., 2021; Xu et al., 2022). However, human activities such as timber and energy extraction, prescribed burning and tourism can also lead to fires (McCarty et al., 2021). Peat fires are a fundamentally different phenomenon to vegetation fires as they burn predominately by smouldering combustion, which is characterised by slow, low temperature and flameless burning with incomplete combustion (Rein, 2013; Huang and Rein, 2017; Rein, 2015). Smouldering is a volumetric phenomenon that also spreads downwards within the

soil (Rein, 2013). Smouldering combustion is heavily influenced by soil properties (Rein, 2015; Archibald et al., 2018) the most important being soil moisture (Rein 2013, Rein 2015). In general, drier soils with deep water tables facilitate greater and deeper burning with high fuel consumption (Purnomo et al., 2020; Lin et al., 2019; Benscoter et al., 2011; Turetsky et al., 2011; Che Azmi et al., 2021). However, fires can still be maintained at moisture contents as high as 160% (Rein, 2013; Hu et al., 2019b; Rein, 2015; Purnomo et al., 2020), indicating the high combustion potential of peatlands. Inorganic content

and bulk density also exert an important control on peat fire ignition and spread (Rein, 2013; Rein, 2015). Higher inorganic content levels in peat, results in slower spread rate of peat fires (Yang and Chen, 2018; Christensen et al., 2020), whilst increased peat bulk density is associated with increased fire spread (Huang and Rein, 2019). Soil temperature is also important, as a peat fire will continue to spread downward and laterally into the soil, existing and spreading underground for months, until it is too cold to maintain a fire (Lin et al., 2021). On average, peat fires burn 12cm deep into the soil (Santoso

et al., 2019), but can burn to as deep as 50cm (Rein, 2015).

Due to the large quantities of carbon sequestered in peatlands, peat fires can release vast amounts of carbon, estimated to be roughly equivalent to 15% of that of anthropogenic emissions (Lasslop et al., 2019; Loisel et al., 2020; Poulter et al., 2006; Rein, 2015). Carbon emissions from peat fires are heavily influenced by the depth of burn, as the deeper a peat fire burns the

larger the pool of carbon that is exposed to combustion (Lin et al., 2021; Huang and Rein, 2017; Che Azmi et al., 2021). Smouldering peat fires also emit a range of gas species, including $CO_2$, $CO$, $CH_4$ and $NH_3$ alongside a suite of aerosols and particulates (Hu et al., 2019a; Voulgarakis and Field, 2015), while they are also dominant in driving the interannual variability of global fire emissions and their consequent effects on global atmospheric composition (van der Werf et al., 2010; Voulgarakis et al., 2015). Aerosols and particulates from peat fires results in degradation of air quality and can lead to

haze events (Turetsky et al., 2015; Hu et al., 2018), consequently disrupting transport, tourism, and agriculture (Hu et al.,



2018; Heil and Goldammer, 2001). Haze also leads to respiratory and cardiovascular problems (WHO, 2006; Hu et al., 2018), with an estimated 25,000 to 50,000 premature deaths due to Arctic wildfire attributed PM2.5 (Silver et al., 2023). For example, in 2010, peat fires surrounding Moscow led to extreme air pollution, and 11,000 additional deaths (Konovalov et al., 2011; Shaponshnikov et al., 2014). Therefore, peat fires are of major concern for the climate and air quality.


Peat fires also have widespread impacts on ecosystems, through altered ecosystem composition and successional trajectories, changes to moisture and nutrient dynamics including increased evapotranspiration, which may alter the functioning of peatlands and further carbon losses (Kettridge et al., 2015; Kettridge et al., 2019; Mekonnen et al., 2021). In the high latitudes 50% of peatlands are affected by permafrost (Hugelius et al., 2020). Following a fire, permafrost can be exposed to

warming resulting in degradation, thermokarst development and further carbon losses (Chen et al., 2021; Nitze et al., 2018). Therefore, peat fires have the potential to cause large shifts in ecosystem functioning and escalate carbon emissions from peatlands.

Peatlands are becoming increasingly vulnerable to fires (York et al., 2020). The Arctic is currently warming at twice the rate

of the global average (Bruhwiler et al., 2021), this alongside decreased precipitation can lead to earlier snow melt and increased water deficits, thus increasing peatland vulnerability to fires and burnt area (Talucci et al., 2022). Land use change, drainage, agriculture, and logging are also increasing peatland vulnerability (Rein, 2015; Langner and Siegert, 2009). Coincidently, lightning frequency has increased substantially in the high latitudes (Veraverbeke et al., 2017), alongside an expansion of human populations and activities into the high latitudes (Bartsch et al., 2021). Increasing lightning and human

ignitions, combined with amplified peatland vulnerability to wildfires is resulting in an increase in fire activity across the high latitudes (McCarty et al., 2021) and risks switching peatlands from fire resistant systems to fire prone systems (Turetsky et al., 2015). For example, 2019, 2020 and 2021 saw the largest fire years on record in North-east Siberia, driven by increased summer temperatures, earlier snow melt and greater plant water stress (Descals et al., 2022; Scholten et al., 2022).

Climate change is expected to amplify the vulnerability of peatlands to wildfires through rising temperatures, increased frequency and intensity of droughts, and increases in fire weather (Thompson et al., 2019; Descals et al., 2022; Lund et al., 2023). Lightning strikes in the high latitudes are expected to increase by 113% by 2100 (Chen et al., 2021). As a result, peat fires are expected to increase in frequency and severity (McCarty et al., 2021; Turetsky et al., 2015). Increased fire occurrence and severity leads to greater carbon emissions from fires and may result in a positive feedback loop with the

climate system, and potentially a catastrophic loss of carbon from the northern high latitudes (Mack et al., 2011; Chen et al., 2021; Turetsky et al., 2015), potentially resulting in peatlands switching from a carbon sink to source by 2100 (Wilkinson et al., 2023).



Despite the importance of peat fires they are currently not explicitly incorporated into most fire models, meaning that the important climate and carbon feedbacks cannot be accurately assessed. At present, the Community Earth System Model (CESM) is the only model to represent peatland burning through its fire / land surface model CLM-Li (Li et al., 2013). The CESM approach was a major step forward, but it is limited by the fact that it does not consider the effects of soil properties on peat fires (Li et al., 2013). Fire models in general do not completely reproduce observed patterns of burnt area (Jones et al., 2022), in particular in the high latitudes. The absence of peat fires is often highlighted as a limiting factor in model ability to reproduce present day burning (Mangeon et al., 2016; Teixeria et al., 2021). Therefore, at present the capacity of fire models to predict future fire activity is limited (Jones et al., 2022). Peat fire representation in models is also key to accurately representing the Northern peatland carbon balance in Earth system models (Wilkinson et al., 2023).

The INteractive Fire and Emission algoRithm for Natural envirOnments (INFERNO) is a reduced complexity fire model, that is part of the Joint UK Land Environment Simulator (JULES) land surface model (Mangeon et al., 2016; Burton et al., 2019). INFERNO estimates plant functional type burnt area, utilising lightning and population density to calculate ignitions and key variables such as relative humidity, precipitation, soil moisture, temperature, and fuel load, to calculate flammability (Mangeon et al., 2016). INFERNO has been shown to accurately diagnose global burnt area compared to observational data from the Global Fire Emissions Database (GFED) (Mangeon et al., 2016), and compares well to other fire models on a global scale (Hantson et al., 2020). However, over the northern high latitudes INFERNO fails to capture a significant amount of burnt area, particularly in Canada, Alaska, and Siberian Russia. INFERNO also underestimates carbon emissions and fails to capture the interannual variability in these emissions (Mangeon et al., 2016). Mangeon et al. (2016) put these underestimates down to the lack of representation of peat fires in INFERNO. We address this gap here by developing a new peat fire parameterisation in INFERNO.

## 2 Model description and developments

INFERNO-peat is a simplified peat fire model, which utilises the existing JULES-INFERNO framework, to add additional burnt area and carbon emissions from peatland burning. At present INFERNO-peat is an offline model run in python vn3.8, using outputs from JULES-INFERNO (detailed below). Figure 1 provides an overview of the model. In summary, INFERNO-peat utilises ignition data from population density and lightning, along with PFT flammability calculated by INFERNO and PFT fractions from JULES to estimate the number of potential peat fire ignitions. The likelihood of those ignitions developing into a peat fire is represented by a parametrisation of peat combustibility based on key relationships with soil moisture, inorganic content, and bulk density (Frandsen 1997). We also parameterised the depth of burn in peatlands using critical soil temperature (Lin et al., 2021) and water table depth, with soil and hydrology simulated by JULES. A parametrisation of peatland carbon emissions was also implemented using calculated peat burnt area, depth of





burn and the carbon content existing in the peat (Lin et al., 2021). In the sub-sections below, we present the individual steps of the peat fire simulation in more detail.

**Figure 1: Schematic summarising the new parameterisations introduced as part of INFERNO-peat, the input variables, and their sources.**

## 2.1 Peat fire ignitions ($I_{peat}$)

The majority of peat fire ignitions result from a pre-existing flaming vegetation fire (Rein, 2013). To account for this, peat fire ignitions are based on the number of flaming vegetation fires in a grid box identified from INFERNO (Equation 1). Here, the third ignition mode of INFERNO is used unchanged from Mangeon et al. (2016). Total ignitions $(I_T)$ are comprised of human ignitions and suppressions, based on population density, and varying natural ignitions, based on cloud-to-ground lightning strikes. The flammability of each of the 13 plant functional types (PFTs) represented in JULES is also calculated using the original equations from Mangeon et al. (2016) in INFERNO. Flammability of each PFT ($Flam_{PFT}$) is based on key climatic drivers of fires such as temperature, relative humidity, and precipitation, alongside fuel density and soil moisture



(Mangeon et al., 2016). From equation (1), the number of peat fires ($I_{peat}$) in a particular PFT is given by the flammability ($Flam_{PFT}$) multiplied by the gridbox ignition rate ($I_T$) and the fraction of the gridbox occupied by that PFT ($Frac_{PFT}$):


$$I_{peat} = \sum_{PFT} I_T \cdot Flam_{PFT} \cdot Frac_{PFT} \qquad (1)$$

### 2.2 Peat combustibility ($Comb_{peat}$)

For each grid box where peat is located, the combustibility of peat is calculated using equation (2) (Frandsen, 1997; Purnomo et al., 2020).

$$Comb_{peat} = \frac{1}{1+\exp(-(B_0+(B_1 \cdot SM)+(B_2 \cdot IC)+(B_3 \cdot \rho)))} \qquad (2)$$

Combustibility describes the probability of a peat fire igniting and spreading. Peat combustibility depends on the peatland soil moisture ($SM$) as a fraction of saturation, inorganic content ($IC$) and bulk density ($\rho$). Soil moisture is the most important variable affecting the ignition and spread of peat fires (Rein, 2013; Rein 2015). Here, soil moisture and peat combustibility exhibit a reverse sigmoid curve, where the likelihood of peat combusting is high at low $SM$ and low at high

$SM$ (Frandsen, 1997; Figure 2a). Fixed values are utilised for $IC$ and $\rho$ of 9.4% and 222kg/m$^3$ respectively (Frandsen, 1997), due to a lack of robust observational datasets of peatland-specific IC and BD on a global scale, and to avoid adding additional sources of uncertainty into the model. B values represent constants identified by Frandsen et al., (1997), where $B_0$ = -19.8198, $B_1$ = -0.1169, $B_2$ = 1.0414, $B_3$ = 0.0782.

### 2.3 Peat burnt area ($BA_{peat}$)

Calculating burnt area from peat fires uses a similar approach to how PFT burnt areas are calculated in JULES (Mangeon et al., 2016). In INFERNO, average burnt area values for each PFT were heuristically determined (Mangeon et al., 2016). However, here we obtain an average peat burnt area ($\overline{BA_{peat}}$) from Santoso et al. (2019), who estimated an average peat fire burnt area, depth, and emitted carbon for boreal peat fires based on reported field studies between 1983 and 2015. For use in INFERNO-peat, anomalously large values reported from Santoso et al. (2019) were omitted from the average. Therefore, an

average peat burnt area of 381.7 was used. The peat burnt area is then calculated following equation (3), where $BA_{peat}$ (fraction of peat burnt during each timestep) results from combining $I_{peat}$, $Comb_{peat}$, $\overline{BA_{peat}}$ and grid box peatland fraction ($Frac_{peat}$).

$$BA_{peat} = I_{peat} \cdot Comb_{peat} \cdot \overline{BA_{peat}} \cdot Frac_{peat} \qquad (3)$$

### 2.4 Depth of burn ($BD_{peat}$)

We adapt the scheme used by Lin et al. (2021) to estimate the depth of burn resulting from a peat fire. Here we estimate the critical soil temperature ($T_{crit}$) which represents the minimum environmental temperature that can sustain a smouldering fire





in peatlands and is driven by the moisture content (*SM*) of the soil (Lin et al., 2021; Equation (4)). Lin et al. (2021), identified a linear relationship between $T_{crit}$ and SM, with increasing SM increasing the $T_{crit}$, which allows dry peat to burn at extremely low temperatures (Figure 2B). INFERNO-peat uses outputs of soil temperature from JULES to then locate at what depth within the soil column $T_{crit}$ is reached. We then assume that a peat fire will burn to this depth, or the depth of the water table (zw) if that is higher.

$$T_{crit} = (42 \times SM) - 28 \qquad (4)$$

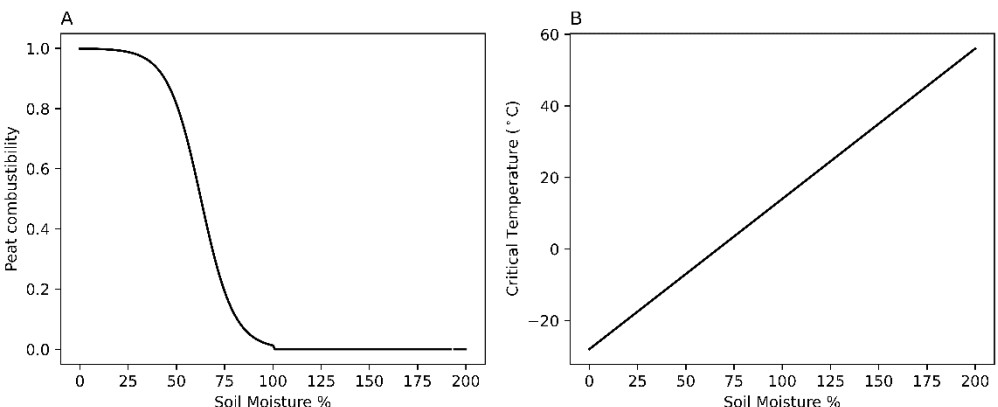

Figure 2: Relationship between soil moisture and A) peat combustibility and B) critical temperature.

## 2.5 Carbon emissions ($C_{peat}$)

Total emitted carbon from a peatland burning is calculated using Equation (5), and reflects common carbon emission calculations (Lin et al., 2021; Che Azmi et al., 2021; Hu et al., 2018). Total emitted carbon from peat fires ($C_{peat}$) is obtained from peat burnt area ($BA_{peat}$) the depth of burn ($BD_{peat}$) and the carbon content of the peat in that grid box (*C*).

$$C_{peat} = BA_{peat} \times BD_{peat} \times C \qquad (5)$$

## 3 Experimental set up and evaluation

INFERNO-peat is run as an offline model using python version 3.8, requiring output files from JULES and INFERNO. JULES-INFERNO outputs, for the majority of variables, were obtained from a TRENDY JULES simulation, at JULES vn5.4 using the TRIFFID dynamic vegetation model. For the peat soil variables (SM, zw, tsoil), an experimental JULES run utilising new peat module developments (Chadburn et al., 2022), was used which assumes from 50° North all soils are organic, therefore producing a better representation of the northern peatland soil physics and dynamics. Standard JULES operates in a similar manner, but instead assumes that all soils are mineral. Within early INFERNO-peat testing this was identified to be causing a systematic bias in the model towards peatlands being drier than they should be, and consequently resulting in inflated burnt area estimates. This was resolved when using organic soil moisture (Supplementary materials S1).



HYDE population density data (Hurtt et al., 2011) was used to calculate human ignitions and suppressions. For natural
ignitions, we ran INFERNO-peat with two datasets. Firstly, as in the original INFERNO (Mangeon et al., 2016), we used a
monthly lightning climatology from LIS-OTD (Lightning Imaging Sensor – Optical Transient Detector) (Christian et al.,
2003). However, a lightning climatology does not represent year to year variation in lightning and therefore could contribute
to inaccuracies in the model. Consequently, we also used the WGLC (WWLLN Global Lightning Climatology and
timeseries) lightning timeseries produced by WWLLN (Worldwide Lightning Location Network) covering 2010-2020 for
comparison. Gridded peatland fractions and their respective carbon contents were prescribed to the model from the Northern
Peatland Dataset (Hugelius et al., 2020).

INFERNO-peat was run at N96 resolution (1.25° latitude X 1.875° longitude) at monthly timesteps from 1997 to 2014 for the
LIS-OTD lightning run and from 2010-2014 for the WGLC lightning run. Due to the availability of data from the organic
soil moisture run, INFERNO-peat could only be run up until 2014.

To evaluate model performance, burnt area data from GFED4s (van der Werf et al., 2017), GFED5 (Chen et al., 2023a; Chen
et al., 2023b) and FireCCILT11 (Otón et al., 2021) was used. The ABoVE-FED (Potter et al., 2022) dataset was also used to
evaluate burnt area in Alaska and Canada. Multiple observational datasets were used for evaluation due to known
deficiencies in the ability of remote sensing based products, in particular MODIS which is used in GFED4s, in being able to
detect peat fires in the high latitudes (McCarty et al., 2021). GFED5 was therefore used as the most up to date product
which, on top of MODIS burned area, utilises high resolution observations from Landsat and Sentinel-2, leading to a 61%
increase in burned area globally compared to GFED4s (Chen et al., 2023). Additionally, FireCCILT11 which is based off
AVHRR images, was used as a non-MODIS based comparison (Otón et al., 2021). Spatial correlations using Pearson's R,
alongside the RMSE metric were used to diagnose spatial performance in burnt area. Temporal correlations were also
conducted for the entire timeseries, alongside comparisons of variation metrics such as standard deviation and the coefficient
of variation. Carbon emissions estimates were evaluated against total carbon emissions from GFED4s (van der Werf et al.,
2017), GFED500m (van Wees et al., 2022) and GFAS (Kaiser et al., 2012). Similarly to burned area, multiple products were
used in addition to GFED4s. GFED500m enhances the accuracy of carbon emissions estimations by increasing the spatial
resolution to 500m, as well as differentiating between aboveground and belowground carbon emissions, allowing for
enhanced analysis of INFERNO-peat's performance (Van Wees et al., 2022). GFAS, which is based on MODIS fire
radiative power (FRP) observations, was used as an alternative to the GFED family of products (Kaiser et al., 2012).
Temporal correlation and variation metrics were again examined for the carbon timeseries results. These analyses were also
carried out on subregions of the high latitudes (Figure S4). All datasets used for evaluation were resampled to N96 grid. A
null model (Standard JULES-INFERNO vn5.4), without peat fires was also used for comparison.





## 4 Results

### 4.1 Burnt area

INFERNO-peat results in an overall improvement compared to the original INFERNO in total BA when evaluated against observations (Figure 3). On average peat fires contribute an additional simulated 0.305 M km$^2$ of burnt area per year across
the high latitudes, bringing the total INFERNO-peat burnt area to within 0.09 M km$^2$ of the GFED5 observations (Table 1). INFERNO-peat allows us to represent clusters of burning more accurately across the high latitudes compared to INFERNO, especially in Western Canada and central and eastern Siberia. However, overestimations are clear in Western Russia and Eastern Canada, whilst in Alaska we are underrepresenting the burning occurring. When driving the lightning ignitions in the model with WGLC lightning timeseries, we capture significantly less burning than when using the LIS-OTD climatology
(Figure 3b, c). Whilst this brings the annual total closer to that seen in FireCCILT11, it results in a large underestimation compared to GFED5 (Table 1). There is a notable area of Southern Russia where there is a high degree of burning in all observational datasets, but which has minimal burning in INFERNO or either of the INFERNO-peat simulations. According to the land cover types modelled by JULES, this area is dominated by C3 crops, and the Northern Peatland Dataset indicates minimal peatland coverage (Figure S3 and S5). Therefore, the underestimations seen in Southern Russia are likely a result of
INFERNO underestimating cropland burning globally rather than representing region-specific agricultural fire management (Burton et al., 2021).

**Table 1: 2010 to 2014 average annual burnt area fraction and statistics for the various models and observations.**

| Model | BA | R GFED4S | GFED5 | FireCCILT11 | RMSE GFED4S | GFED5 | FireCCILT11 |
|---|---|---|---|---|---|---|---|
| INFERNO | 0.215 | 0.485 | 0.530 | 0.486 | 0.061 | 0.095 | 0.068 |
| INFERNO-peat LIS-OTD | 0.520 | 0.414 | 0.431 | 0.398 | 0.098 | 0.124 | 0.099 |
| INFERNO-peat WGLC | 0.282 | 0.475 | 0.510 | 0.467 | 0.071 | 0.099 | 0.076 |
| GFED4s | 0.172 | | 0.820 | 0.868 | | 0.068 | 0.028 |
| GFED5 | 0.429 | 0.820 | | 0.879 | 0.068 | | 0.053 |
| FireCCILT11 | 0.276 | 0.868 | 0.879 | | 0.028 | 0.053 | |





**Figure 3: 2010 to 2014 average annual burnt area fraction for INFERNO (a), INFERNO-peat driven by LIS-OTD climatology (b), INFERNO-peat driven by WGLC timeseries (c), GFED4s (d), GFED5 (e) and fireCCILT11 (f).**

Over the entire model run (1997 to 2014), there are large improvements in the representation of interannual variability (IAV) in burnt area in INFERNO-peat compared to INFERNO (Figure 4). We capture significantly higher IAV in INFERNO-peat, with the standard deviation increased from 0.011 in INFERNO to 0.041 in INFERNO-peat, bringing the model much closer to the magnitude of IAV seen in the observations (Table 2). Furthermore, there is an improvement in the R value in INFERNO-peat across all observational datasets, meaning that we are also more accurately capturing the timing of the IAV in burnt area (Table 2). However, when compared to GFED5 there is still a noteworthy underestimation of burnt area IAV in INFERNO-peat, mainly between 2001 and 2011. In particular, the large spikes in burning occurring in 2003 and 2008 are not as pronounced in INFERNO-peat, even though they are captured in a qualitative sense.





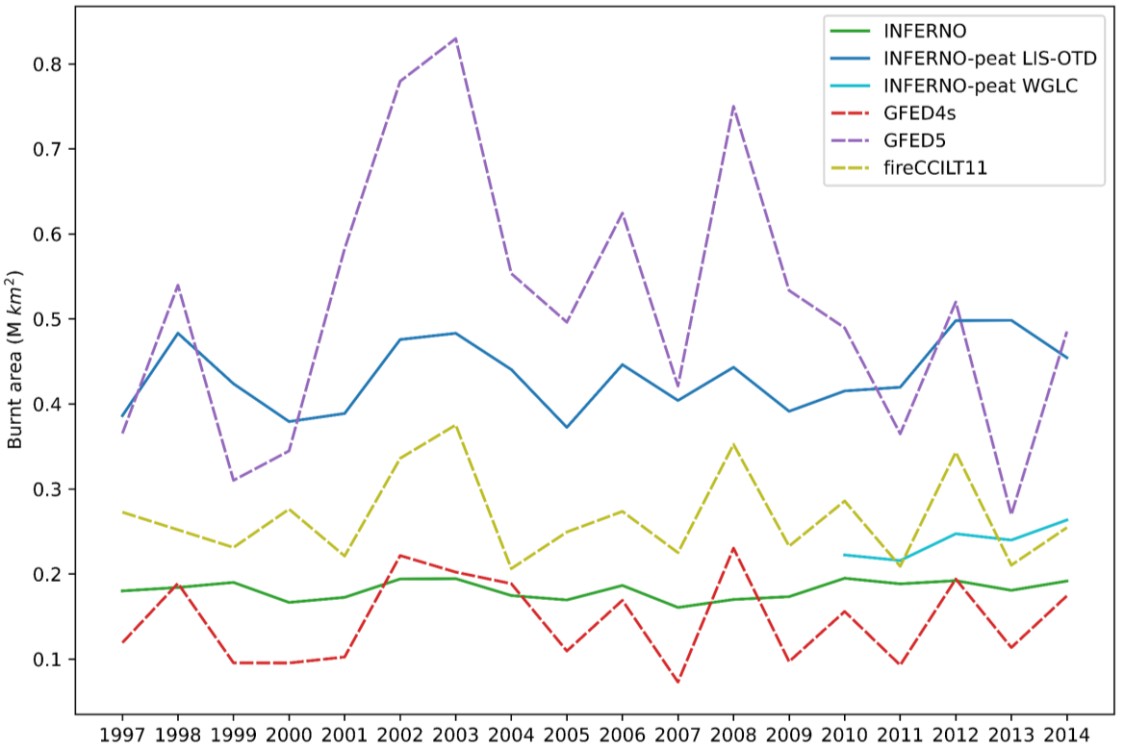


**Figure 4: The annual total burnt area across the high latitudes (<50º N) from 1997 to 2014 compared between the models (indicated by solid lines) and observations (indicated by dashed lines).**

**Table 2: 1997 to 2014 average annual burnt area, temporal standard deviation (std), coefficient of variation (cv) and temporal correlation (R), for models and observations. Correlation coefficients for the WGLC period (2010-2014) are**
**available in table S1.**

| Model | Average annual BA (M km$^2$) | std | cv | R (GFED4s) | R (GFED5) | R (fireCCILT11) |
|---|---|---|---|---|---|---|
| INFERNO | 0.181 | 0.011 | 0.058 | 0.463 | 0.206 | 0.370 |
| INFERNO-peat LIS-OTD | 0.434 | 0.041 | 0.094 | 0.676 | 0.323 | 0.394 |
| INFERNO-peat WGLC | 0.238 | 0.017 | 0.072 | | | |
| GFED4s | 0.146 | 0.050 | 0.340 | | 0.759 | 0.709 |
| GFED5 | 0.514 | 0.155 | 0.300 | 0.759 | | 0.692 |
| fireCCILT11 | 0.267 | 0.051 | 0.192 | 0.709 | 0.692 | |

Regional features and patterns are evident across the high latitudes. To evaluate this, seven subregions were examined (Figure S4). Across all three North American subregions INFERNO-peat overestimates burning compared to the observations including the North America only dataset ABoVE-FED (Figure 5). Whilst this is only minor in Alaska, the
overestimation is particularly pronounced in Western Canada, with burnt area in INFERNO-peat around almost 4 times greater than in GFED5. However, in all other subregions the opposite is true, especially when compared to GFED5. In





central Russia INFERNO-peat burns on average 0.151 M km² per year which, significantly lower than the 0.187 M observed in GFED5. Similarly, we also see a large underestimation in burning in Eastern Russia, when compared to GFED5 and FireCCILT11. Despite this underestimation we do still see an improvement in RMSE in this region, which is not seen in others (Table S2). In accordance with the overall high latitudes results, INFERNO-peat captures more interannual variability in all regions compared to INFERNO (Table S2 and S3). This increase is a lot more pronounced in western Canada, eastern Canada, central Russia, and eastern Russia, which are also the regions where we see the greatest change between the models.

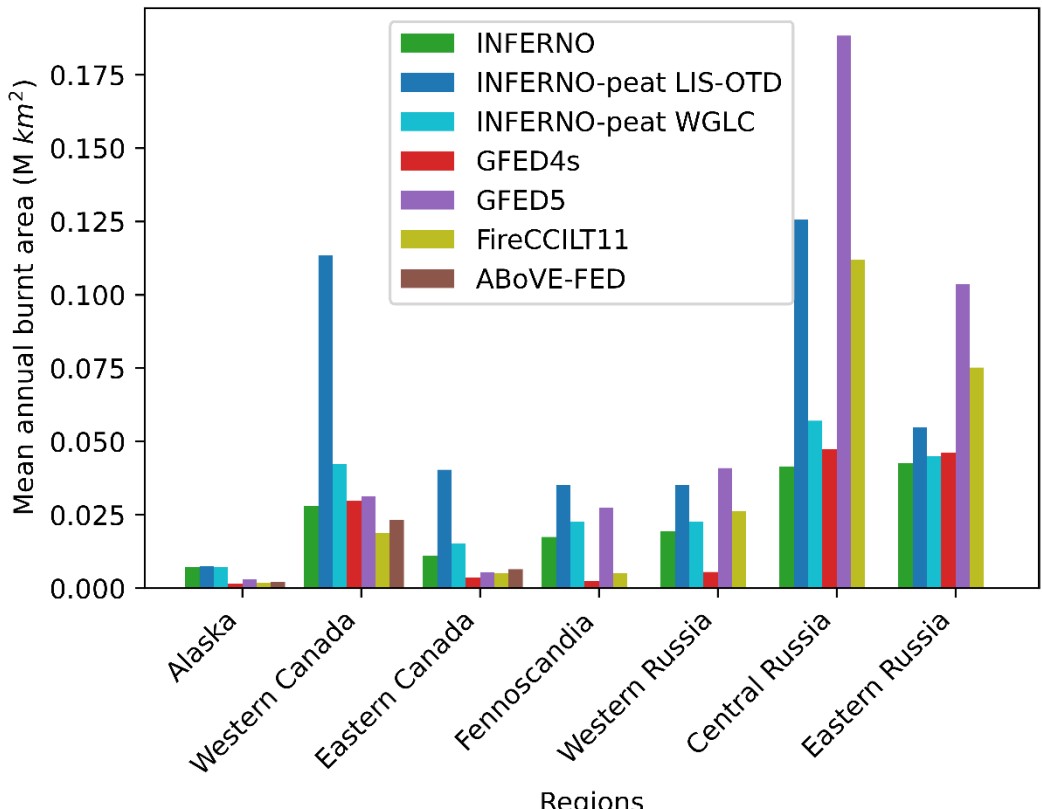

**Figure 5: The average annual burnt area (2010-2014), from INFERNO, and INFERNO-peat driven by the LIS-OTD and WGLC lightning data, compared to the observations, in each subregion.**

### 4.2 Carbon emissions

On average, peat fires emit an additional 224.10 Tg of carbon per year in INFERNO-peat (Figure 6), significantly more than the 103.28 Tg modelled by INFERNO. This brings emissions estimates closer to the 305.35 Tg C in GFED500m, and 248.57 Tg C in GFAS (Table 3). Not only are annual averages brought closer to the observations, but INFERNO-peat also allows for more accurate representation in the interannual variability in carbon emissions. We also see an improved temporal correlation between INFERNO-peat and the GFED4s and GFED 500m observations over this time period when compared to INFERNO.





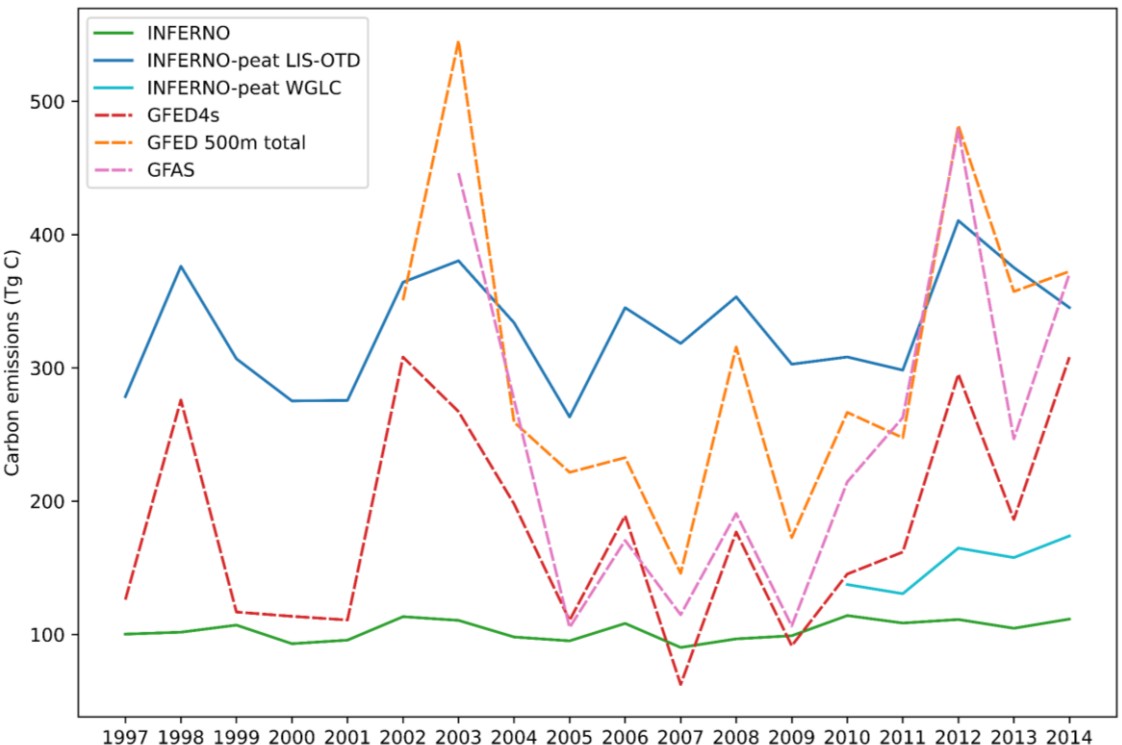

**Figure 6: The annual total carbon emissions from fires across the high latitudes (<50º N) from 1997 to 2014 compared between the models (indicated by solid lines) and observations (indicated by dashed lines).**

The GFED 500m product is uniquely useful as it has a differentiation between aboveground and belowground carbon emissions. Belowground emissions come from the burning of organic matter within the soil, which occurs predominantly during peat burning. When compared to GFED 500m, we are representing fire emissions and their sources well. Specifically, Figure 7 shows the breakdown in above and below ground emissions, and peat vs non-peat emissions from INFERNO-peat. We can see that INFERNO, which only represents vegetation fires, does an adequate job at capturing the aboveground burning, albeit not capturing the interannual fluctuations in burning. Emissions from peat fires from INFERNO-peat fall relatively in-line with belowground burning in GFED 500m. This indicates that the changes we have implemented are successfully capturing the observed below ground emissions.





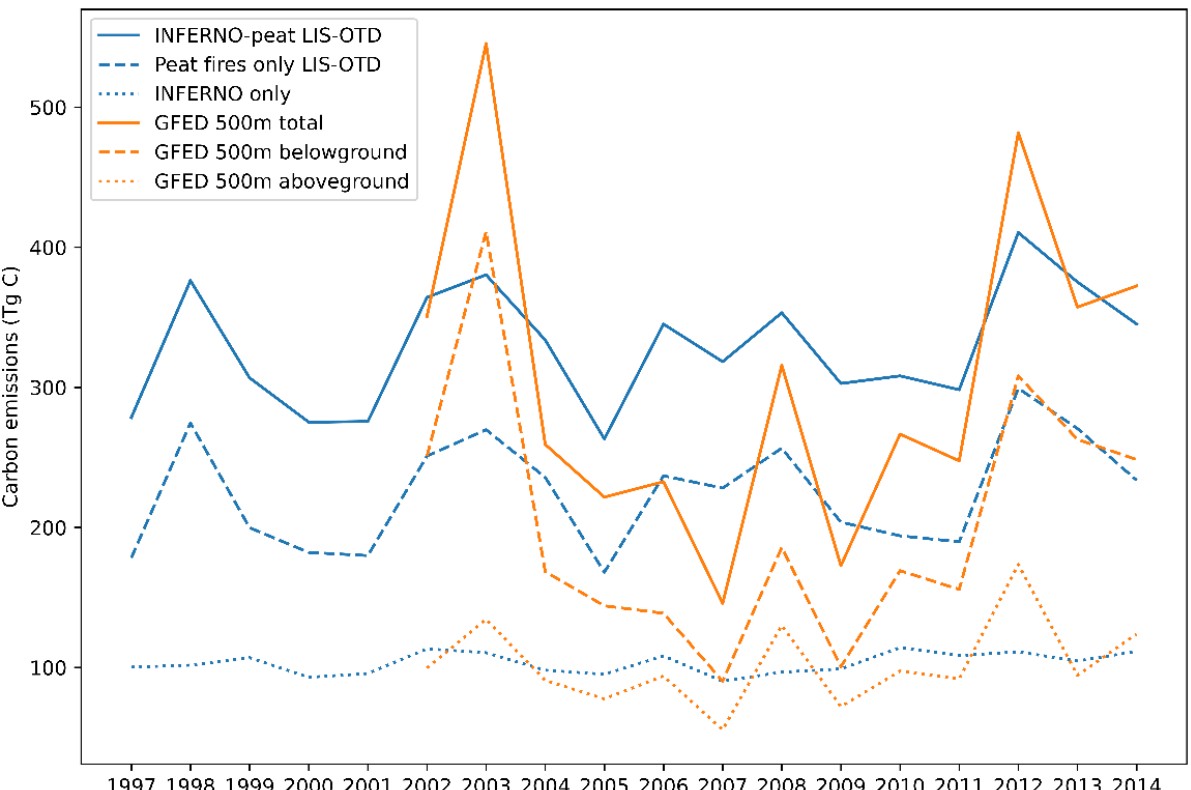

**Figure 7: The annual total carbon emissions from fires across the high latitudes (<50° N) from 1997 to 2014 compared between the GFED 500m products above and belowground burning, and INFERNO-peat emissions split to peat fires only and INFERNO only.**

**Table 3: Timeseries annual carbon emissions, standard deviation (std), coefficient of variation (cv) and temporal correlation R coefficients for models and observations over 2003 to 2014.**

| Model | Average annual C emissions (Tg) | std | cv | R (GFED4s) | R (GFED500m) | R (GFAS) |
|---|---|---|---|---|---|---|
| INFERNO | 103.282 | 7.341 | 0.071 | 0.685 | 0.601 | 0.670 |
| INFERNO-peat LIS-OTD | 328.377 | 41.719 | 0.127 | 0.735 | 0.790 | 0.755 |
| INFERNO-peat WGLC | 152.873 | 16.390 | 0.107 | | | |
| Peat only LIS-OTD | 225.095 | 38.498 | 0.171 | | | |
| GFED4s | 180.192 | 77.259 | 0.429 | | 0.858 | 0.914 |
| GFED500m - total | 305.350 | 111.106 | 0.364 | 0.858 | | 0.912 |
| GFED500m - aboveground | 102.718 | 29.674 | 0.289 | | | |
| GFED500m - belowground | 202.632 | 86.844 | 0.429 | | | |
| GFAS | 248.574 | 121.469 | 0.489 | 0.914 | 0.912 | |




The carbon emissions modelled by INFERNO-peat vary greatly between subregions (Figure 8, Table S4, S5). For example, whilst burnt area total is very close to the observations in Alaska (Figure 5), carbon emissions are underestimated. There is an even more pronounced underestimation in carbon emissions in Eastern Russia, where emissions captured by the GFED 500m product are over 6 times greater than the emissions modelled by INFERNO-peat. However, INFERNO-peat is overestimating carbon emissions by over double in western Canada and central Russia compared to GFED 500m and GFAS.

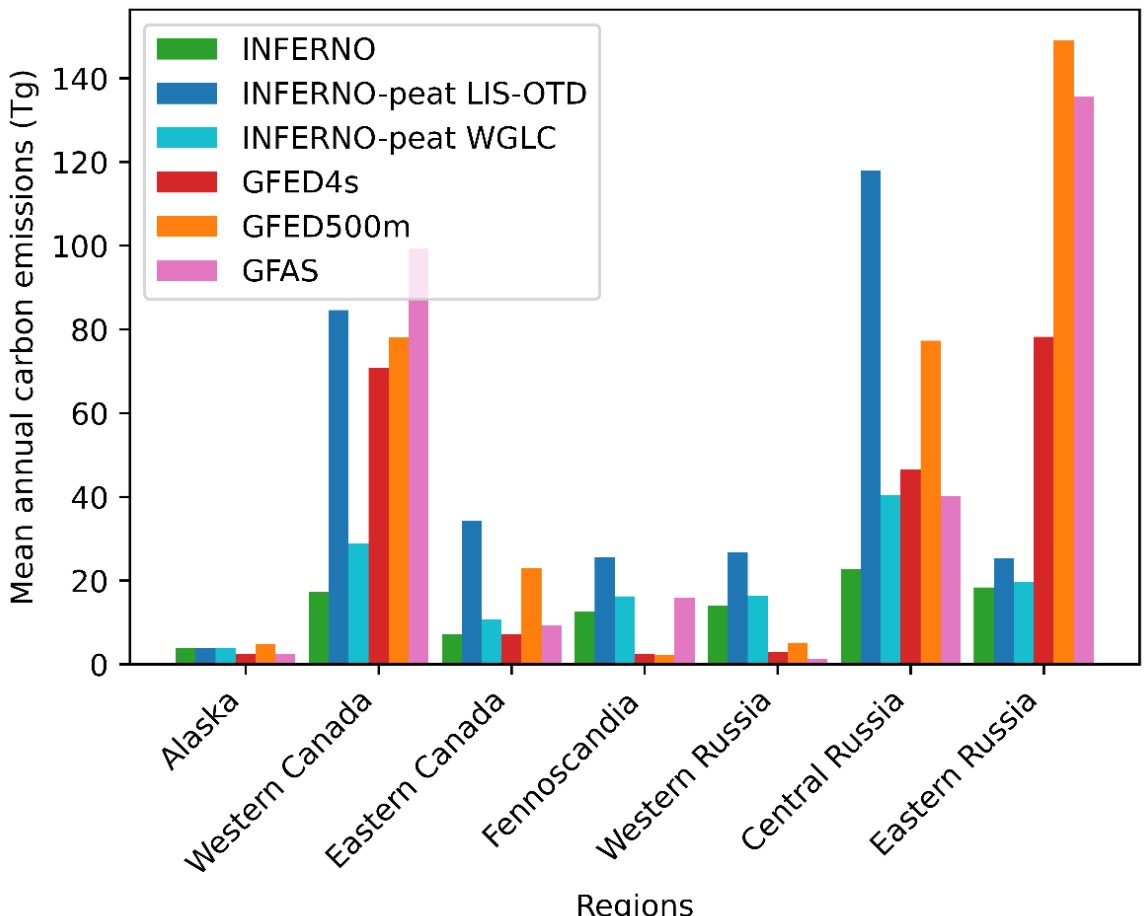


**Figure 8: The average annual carbon emissions (2010-2014), from INFERNO, and INFERNO-peat driven by the LIS-OTD and WGLC lightning data, compared to the observations, in each subregion.**

By examining the aboveground and belowground burning reported in the GFED500m product, we can look at what is potentially driving these differences (Figure 9). For example, in Alaska INFERNO is capturing the majority of GFED500m 310 aboveground emissions, but emissions from the peat model are negligible. This means that the deficit in carbon emissions estimate in Alaska is likely a result of not capturing the belowground burning happening in GFED500m. Similarly in eastern Russia, peat fires modelled by INFERNO-peat emit on average only 6.983 Tg C per year, as opposed to the 107.002 Tg of carbon observed from belowground burning in GFED500m. In contrast in western Canada and central Russia, carbon





emissions are over double that from peat fires as they are from belowground burning in GFED500m, meaning that

INFERNO-peat is likely overrepresenting peat burning in these regions.

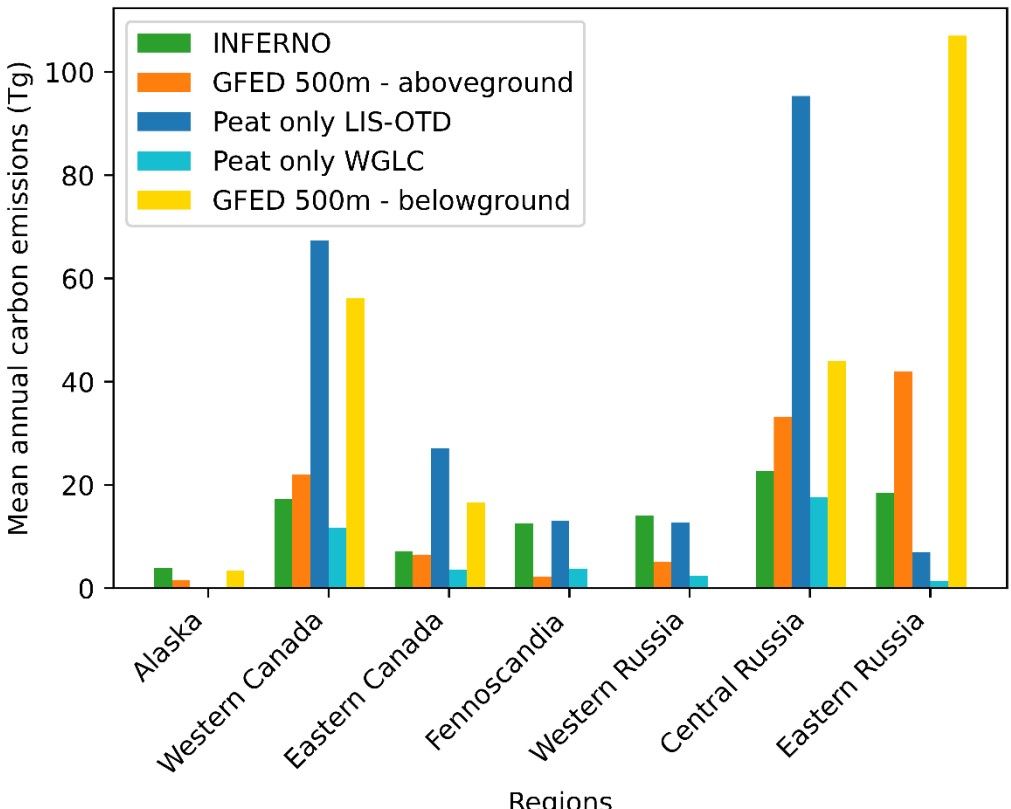

**Figure 9: The average annual carbon emissions (1997-2014), from INFERNO, and peat fires only from INFERNO-peat driven by the LIS-OTD and WGLC lightning data, compared to the aboveground and belowground burning from the GFED 500m product, in each subregion.**

Underestimations in burnt area and carbon emissions seen in Eastern Russia and Alaska may be a result of inherent biases within JULES-INFERNO. In these regions, temperatures are low, which results in INFERNO underestimating the flammability of vegetation in these areas, which causes INFERNO-peat to underestimate the number of ignitions in these areas, resulting in underestimations in burnt area and emitted carbon. Furthermore, simulated vegetation bias in the TRIFFID dynamic vegetation model within JULES, results in a high proportion of the land surface being covered by non-vegetative

surface types, predominantly bare soil (~67% of the land surface north of 60 degrees is classed as non-vegetative) (Figure S5). Therefore, the number of possible ignition events in INFERNO-peat is further reduced. However, the opposite may be true in central Russia and Western Canada where we see large overestimations in burnt area and carbon emissions. In these regions the most common plant functional types are needleleaf deciduous trees, whereas, in reality, these regions contain large amounts of herbaceous wetlands, a PFT which is not currently represented in JULES. This may be contributing to a



possible inflation in the number of flaming vegetation fires in these regions, leading to overestimates in burnt area and consequently carbon emissions in INFERNO-peat.

## 5 Discussion

Through the explicit representation of peat fires in INFERNO-peat, we have improved INFERNO's ability to capture burning in the northern high latitudes, and in particular improved simulated estimates of wildfire carbon emissions and their
interannual variability. According to INFERNO-peat, peat fires accounted for, on average, 58% of burned area, and 68% of carbon emissions north of 50 degrees latitude, therefore peat fires have a large impact on simulated model performance. At present the only other fire model that represents peat burning is CLM4 (Li et al., 2013). However, the inclusion of peat burning in CLM4 did not show substantial improvements in the simulation of fire in the high latitudes of North America and Eastern Siberia (Li et al., 2013). This was attributed to a wet simulation bias seen in CLM4, whereby the latent heat flux was
underestimated leading to an inflation in the amount of water held by the land (Li et al., 2013). This is in contrast to our findings, where we see a substantial increase in burnt area, particularly in North America, presumably as a result of using organic soil moisture, allowing us to represent the hydrology of peatlands more accurately. However, in Eastern Siberia, whilst INFERNO-peat does improve our estimations of burning, we still see much less burning compared to observations. Li et al. (2013) attributed a similar feature they found to low fuel loads in their model. Similarly, land cover fractions modelled
by the TRIFFID dynamic vegetation model in JULES show low levels of vegetation in Eastern Siberia, and in particular show a dominance of bare soil (Figure S5). In reality, much of these tundra ecosystems in Eastern Siberia are dominated by grass, moss and lichen. At present there is no moss or lichen PFT in JULES, and therefore the amount of burnable area is significantly underestimated. Improvements in the simulation of high latitude ecosystems within dynamic vegetation models are vital to improving fire modelling in these regions.


There is significant regional variation in the performance of INFERNO-peat. One notable area of improvement is seen in the Northwest Territories region of Canada, where previously INFERNO has struggled to replicate observed patterns of burning. Simulating fires accurately in this region is of vital importance because they have been shown to have major climatic effects. For example, Canadian fires in 2013 led to high levels of black carbon deposition over Greenland, which resulted in a
lowering of albedo, and consequently a warming effect on the climate (Thomas et al., 2017). Similar improvements are also evident in Russia, where large peat fires are common. For example, fires in 2010 burned at least 40,000 ha of peatlands in the Moscow region (Sirin and Medvedeva, 2022). As shown here, peat fires can lead to large carbon emissions, for example during the summer of 1998, peat and boreal forest fires in Russia burnt 11 million hectares and emitted 176 Tg of carbon (Kajii et al., 2002). Representing a substantial effect on the atmosphere, as well as peatland carbon stores.




However, INFERNO-peat also overestimates peat burning compared to observations in Canada and Fennoscandia. One potential cause of these overestimates may be a result of how humans are represented in INFERNO. At present most global fire models rely on simplistic relationships with population density, like INFERNO, or GDP, which fail to capture the highly complex relationship between humans and fire (Perkins et al., 2022; Teckentrup et al., 2019). Humans use fire as a land
management tool around the world, altering fuel loads, fragmenting landscapes and converting land to agriculture, pasture, or industry (Smith et al., 2022; Perkins et al., 2022; Archibald, 2016). Therefore, improvements are required in how we represent humans within INFERNO, potentially through recent approaches that have accounted for the Human Development Index (Teixeira et al., 2023) or on agent-based modelling (Perkins et al., 2022). Recent analysis has also shown land fragmentation metrics such as road density to exert a strong control on burnt area globally (Haas et al., 2022), and therefore
represents a potential future improvements that could be made to INFERNO.

A lack of interannual variability is a common deficiency across global fire models, with most models considered in FireMIP failing to simulate the interannual variability in fires (Li et al., 2019). However, through the inclusion of peat burning in INFERNO we see substantial increases in interannual variability of burnt area and carbon emissions in the northern high
latitudes. Furthermore, we also see improvements in our ability to capture large fire years, for example in 1998, 2003 and 2012 (Figure 6). Whilst INFENRO-peat cannot replicate real life fire events and only relies on an estimate of the likelihood of burning, the simulated spikes in burning are also seen in observational data and coincide with record fire years. For example, until recently, 2003 was the largest fire season on record in Siberia burning 22 million hectares of land an emitting at least 72 Tg of CO (Talucci et al., 2022). Not only did the 2003 fire season have a major impact on Siberian populations
and ecosystems, but haze and smoke plumes were reported to have reached Japan and the USA (Huang et al., 2009). Similarly in 2012, there were a reported 17,000 wildfires in July and August in Siberia, emitting 48 Tg of CO, with smoke reaching the Pacific Northwest of the USA (Teakles et al., 2017). Consequently, being able to accurately simulate these large-scale fire events and their subsequent emissions is highly important in the assessment of both local and global impacts of wildfires. Furthermore, climate change is anticipated to further increase peat fires, with significant impacts on climate, air
quality and the peatland carbon store (Mack et al., 2011; Chen et al., 2021; Turetsky et al., 2015). Therefore, it is important that fire models represent peat fires in order to better anticipate future changes in burning across the high latitudes over the coming century.

A major challenge in modelling peat burning stems from an absence of robust observational datasets on peat fires, making
model evaluation difficult. MODIS satellite products form the basis of the GFED4s data which is the most used observational dataset to evaluate fire model performance. However, MODIS, and other satellite-based products, likely omit a large number of peat fires, as such fires tend to burn below ground and at low temperatures making them difficult to detect by remote sensing (McCarty et al., 2021). For example, MODIS was shown to be insufficient to detect the peat fires that occurred in the Moscow region in 2010 (Sirin and Medvedeva et al., 2022). Further afield, burnt area estimated from



Sentinel-2 over sub-Saharan Africa, was estimated to be 80% larger than MODIS, through improved detection of small fires (Roteta et al., 2019). If that pattern holds true for the high latitudes, then MODIS-based products could be failing to capture a substantial amount of burning, and consequently carbon emissions. In this study, we have also utilised data from the ABoVE-FED dataset for analysis of North America, which uses the difference normalised burn ratio calculated from Landsat imagery to enhance MODIS estimates (Potter et al., 2022), which provides a slightly more accurate estimation of burning across Canada and Alaska. However, a similar more finely detailed satellite-based product does not currently exist for Russia, nor is there a ground-based dataset available (McCarty et al., 2021). Therefore, it is challenging to sufficiently evaluate the performance of INFERNO-peat due to an underrepresenting of peat fires in observational datasets.

Refinements and developments to INFERNO-peat could further improve the model's capabilities to capture peat burning and the associated emissions. For example, through the use of peat-specific emissions factors (Hu et al., 2018), the modelling scope could be extended to include emissions of other species such as $CO_2$ $CH_4$ and NOx. This would allow for further investigation into the air quality impacts of peat fires, and in particular the impacts on human health. There is also the possibility of the INFERNO-peat scheme being extended globally, with a particular focus on the tropics. Tropical peat fires release vast quantities of carbon to the atmosphere, for example, the 1997 peat fires in Indonesia released up to 2.57 Gt of carbon, equivalent to 40% of the global annual carbon emissions from fossil fuel burning at that time (Page et al., 2002). Therefore, representing peatland burning in these regions, would represent another substantial step forward in fire modelling capabilities. INFERNO-peat can also be used to model future changes in peatland burning over the coming century under different climate change scenarios, improving our ability to anticipate future fire regimes.

## 6 Conclusion

The explicit representation of peat fires in INFERNO-peat improves simulated burnt area estimates and has increased our ability to capture the interannual variability in carbon emissions across the northern high latitudes. Results presented here not only have addressed noted deficiencies in the INFERNO fire model (Mangeon et al., 2016), but also highlight the crucial need for representing peat burning in fire models in order to simulate the release of vast amounts of long-term stored carbon to the atmosphere. The high latitudes are warming at twice the rate of the global average (Bruhwiler et al., 2021), and continued climate change is expected to cause peatlands to dry out, lightning strikes to increase, and the frequency and severity of wildfires to escalate (Talucci et al., 2022; McCarty et al., 2021; Turetsky et al., 2015; Chen et al., 2021). Increases in peat fire frequency and severity, may amplify carbon loss, and create a positive feedback loop on the climate system (Rein, 2013; Hu et al., 2018; Turetsky et al., 2015), ultimately shifting peatlands from sinks to sources of carbon by the end of the century (Swindles et al., 2019; Turetsky et al., 2015; Loisel et al., 2020). Therefore, it is vitally important that fire models include a specific parameterisation of peat fires in order to be able to replicate historical and present day burning

more accurately, allowing for more comprehensive assessments on the impacts of fire on the climate system, air quality and the carbon cycle, both now and in the future.

**Code availability**

The JULES code, used for generating inputs into INFENRO-peat, is freely available from the JULES trunk version 5.4

onwards at: https://code.metoffice.gov.uk/trac/jules (registration required; last access: 12/10/2023). Code for INFERNO-peat is available from: https://doi.org/10.5281/zenodo.10007362

**Author contributions**

KRB developed the parameterisation of peat fires in INFERNO and undertook model analysis and evaluation with help and advice from MK, CB, CP and AV. CB provided the modelled outputs from JULES-INFERNO. EB provided modelled

outputs from JULES organic soil moisture runs. KRB drafted the text and figures, with MK, CB and AV reviewing drafts and contributing to the text.

**Competing interests**

The authors declare that they have no conflict of interest.

**Acknowledgements**

KRB, MK, CP and AV were supported by the Leverhulme Trust through the Leverhulme Centre for Wildfires, Environment and Society, grant number RC-2018-023. CB was funded by the Met Office Climate Science for Service Partnership (CSSP) Brazil project, which is supported by the Department for Science, Innovation & Technology (DSIT). CP acknowledges support from the LEMONTREE (Land Ecosystem Models based on New Theory, obseRvations and ExperimEnts) project, funded through the generosity of Eric and Wendy Schmidt by recommendation of the Schmidt Futures programme. AV has

also been supported by the AXA Research Fund (project title 'AXA Chair in Wildfires and Climate') and received funding from the Hellenic Foundation for Research and Innovation (HFRI) and the General Secretariat for Research and Technology (GSRT), under grant agreement No 3453. The authors would like to thank Yang Chen and James Randerson for providing access to the GFED5 data.



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
