# Peer review of "INFERNO-peat v1.0.0: A representation of northern high latitude peat fires in the JULES-INFERNO global fire model"

_EGUsphere, 2023_

## Author Response (AR1)

Authors responses to referee comments are shown in blue. Changes to the manuscript are in purple.

Reply to referee 1 review

We would like to thank the referee for taking the time to read the manuscript and share their detailed and constructive comments. All their comments and suggestions have been addressed and have been thoroughly beneficial in improving the manuscript.

The authors present an enhancement to the JULES fire model INFERNO, which enables that model to simulate peat fire (ground fire) in northern latitudes. INFERNO-peat is implemented in Python as an external model that is driven by output from JULES-INFERNO. Simulations driven by two different ignitions forcing datasets for 1997 - 2014 were compared to several global and one regional observational datasets. The INFERNO-peat implementation increases northern latitude burnt area estimates bringing it closer to observation products such as GFED5 and FireCCILT11. The spatial correlation with observations is reasonable, but with some biases. The simulations also had larger and more realistic intern annual variability than INFERNO. Emissions were correspondingly increased with INFERNO-peat, closer to total emissions estimates. Again, regional biases were present. The GFED 500 meter product, which contains aboveground and below-ground emission estimates was used to look at emissions more closely. This revealed that biases were dominated by below-ground carbon emissions being too low or too high in different regions. Inaccuracies in estimates of vegetation in JULES-INFERNO are likely propagating through to drive some of the biases in INFERNO-peat as well. Improving prediction of peat fires in models is important because of the both climate and societal impacts. Improvements to INFERNO-peat could come from better representing human behavior and the specifics of peat emissions. The authors also highlight the need for better detection and estimation of ground fires, which is not always observable with current satellites.

The authors have presented a well written and interesting article documenting INFERNO-peat. The model extension is welcome advancement and the simulations results are compelling. This work is relevant and novel. I can only suggest a few minor changes and areas for improvement as follows:

Thank you for the detailed summary of the manuscript, and your kind comments.

In Figure 1. the 'r' is missing in 'moisture'.

Thank you for spotting this mistake. This has been rectified in the revised manuscript.

Figure 1, line 133, has been updated.

The equations in section 2 would be easier to understand if units were given with the first mention of each variable. While the units for some are implied or inferable, others are not. Specifically, I could not determine the units for FlamPFT in section 2.1, soil moisture (SM) in 2.2, and burnt area (BApeat) in section 2.3.

Agreed, the units for all variables have been added into the revised manuscript to aid understanding and avoid any incorrect assumptions.

Units have been added at the first mention of each new variable in lines 143, 155, 169, 174, 175.

In figure 3 the color scale makes it very hard to make out any signal in Fennoscandia and Alaska for most panels. Changing the scale so that true zero cells have their own color (white?) while low non-zero values have another color would make comparisons easier.

Agreed. The scale has been adjusted in the revised manuscript so that low levels of burning are now much more visible.

The colour scale in Figure 3, line 264, has been updated.

Figure 9 is interpretable with aid of the text. However, it would be clearer if the INFERNO bar was marked as "aboveground" and the peat only as "below-ground" in the legend.

Thank you for this feedback, the legend of Figure 9 has been updated in the revised manuscript.

Figure 9, line 350, has been updated to have the INFERNO bars labelled as above and belowground.

The discussion is quite complete. However, a few lines discussing any potential benefits of (or obstacles to) full integration of the INFERNO-peat logic into JULES-INFERNO would be appreciated.

Thank you. Agreed, adding in a discussion of how INFERNO-peat can be integrated into JULES-INFERNO would add extra depth and detail to the manuscript. This has been added into the revised manuscript lines 429 to 432.

Lines 429-432 have been added to the manuscript.

Given the number of colors in supplemental figure 5 it would be helpful to either mark any PFTs not shown in the plot or remove them entirely from the legend.

Thank you for this feedback, the legend is likely too overcrowded. Therefore, in the revised manuscript any PFTs not shown in the plot have been removed entirely from the legend.

The legend of supplementary materials figure S5 has been updated.

Referee 2 – authors response

We would like to thank the referee for taking the time to read the manuscript and share their detailed and constructive comments. All their comments and suggestions have been addressed and have been thoroughly beneficial in improving the manuscript.

This work developed a new model INFERNO-peat. It has some improvements from the original INFERNO model in terms of estimations of burnt area, carbon emissions, etc, especially in northern high latitudes. The major comments are summarized as follows:

Thank you for your clear summary of the manuscript.

How did the model consider the effects of wind speed and ambient temperatures? I believe they play important roles in the ignition and spread of peat fires.

Ambient temperatures are considered as part of INFERNO in calculating PFT flammability. Therefore, whilst INFERNO-peat does not directly use ambient temperature, it is considered in how we calculate the number of peat fire ignitions. Wind speed, although plays a crucial role in the spread of fires, is not included in INFERNO-peat or INFERNO. This is because we do not model individual fire events, nor do we model the spread of fires, and instead try to capture the overall coarse scale patterns in burning, which is more of relevance to global climate/Earth system model applications. We refer to the original INFERNO paper for more information (Mangeon et al., 2016 doi:10.5194/gmd-9-2685-2016).

No changes in manuscript.

L55, "but can burn to as deep as 50cm". Some recent lab experiments showed it can burn at 100 cm depth (Qin et al. 2022)

Thank you for bringing this interesting study to my attention. L55 has now been adjusted to reflect what is being found in lab experiments.

L55 has been amended.

L150, from Eq. 2, the combustibility of peat soil only depends on its moisture content. Even though the authors state MC plays dominate role, there are many studies emphasizing the significance of other factors like inorganic content (Frandsen 1997), ambient temperature, fuel density, etc.

Agreed, whilst soil moisture is often cited as the most important driver of peat fire ignition and spread, factors such as inorganic content and bulk density also play important roles. This is why we also use inorganic content and bulk density in equation 2. As stated in L156, we used fixed values for these variables from Frandsen (1997), due to a lack of datasets on these variables for the high latitudes. Additional factors appeared to not be as well studied and supported as the three used in this manuscript and therefore, were not included at this time to avoid adding additional sources of uncertainty. Future developments of INFERNO-peat could benefit from adding in additional variables.

No changes in manuscript.

Line 165, the unit is missing.

Thank you for spotting, the unit has now been added.

Unit for average peat depth of burn has been added in the revised manuscript, now line 168.

Line 180, the carbon emission calculation is too rough. I understand the authors try to calculate the total emitted carbon. But your cited works either use emission factors or carbon emission flux (7.1 kg C/m2). Assuming that all carbon (C) from the burned fuel (Eq. 5) is completely converted to emissions is far from realistic.

Agreed, as it stands the carbon emissions calculation likely results in an overestimation in the amount of carbon emissions. Initially we chose to use burn depth to determine the amount of carbon pool to burn, rather than a fixed carbon emissions flux to be able to capture the variations in peat fire emissions from fires that burn deep into the soil vs those that don't. However, upon receiving your comment, we decided to implement a fixed value for combustion completeness in the model. We tested multiple model runs using 4 different combustion completeness values based off the surrounding literature. A sensitivity analysis showed that 0.8 was the optimum value to use. A full explanation of this can be seen in the revised manuscript L195-201, and in the revised supplementary materials S2.

The carbon emissions calculation has been updated in the model to include combustion completeness. Lines 195 to 201 in the revised manuscript have been added to explain this new addition and the reasoning behind it. The addition of this new parameter also came along side an addition of a maximum depth of burn to produce more reliable modelled burn depth and consequent carbon emissions. Lines 179 to 187 detail the addition of maximum depth of burn and the reasoning for it.

We carried out a sensitivity analysis to determine combustion completeness and maximum burn depth parameters. The details of this have been added in the supplementary materials section S2.

As a result of these changes the carbon emissions results (section 4.2) are minorly different in the revised manuscript. Table 3 and Figures 6-9 have therefore been revised in the manuscript and Tables S4 and S5 and Figures S9 and S10 in the revised supplementary materials. The changes the addition of combustion completeness and maximum depth of burn only result in small changes in our results, and the conclusions remain the same.

In Fig. 2a, it indicates peat becomes incombustible when MC =100%. But in Line 48, it states "However, fires can still be maintained at moisture contents as high as 160% (Rein, 2013; Hu et al., 2019b; Rein, 2015; Purnomo et al., 2020)". It is because the critical MC can change with other parameter (Frandsen 1997)

This is correct. So, whilst it is possible for peat to burn at 160% moisture content, this depends on other parameters such as inorganic content and bulk density of the soil. In Fig. 2a the combustibility of peat is calculated taking into account soil moisture but also using fixed values of inorganic content and bulk density, thereby altering the critical MC. Clarification on this point has been added into the revised manuscript (L49).

An addition to line 49 has been added.

3, It seems INFERNO-peat can capture more fires in high latitudes but less fires in low latitudes (compared with GFED and fireCCILT11), especially in Eurasia area. Can the authors explain why?

In the lower latitude regions we have studied we do indeed see that INFERNO-peat doesn't capture much additional burning, especially in Eurasia. Simply, this is because there is not much peat in these areas (please see Supplementary Figure 3), therefore we would not expect that adding peat fires into the model would improve model performance in these specific areas. Furthermore, these areas in the model are largely dominated by C3 crops, suggesting that the underestimations seen in these regions are a result of INFERNO underestimating cropland burning. This is detailed in lines 236-241 in the manuscript.

No changes to manuscript.

5, 8, 9: The authors compare the average values over several years. However, providing subplots with the average values for each region on a yearly basis would be more compelling.

We chose to use these plots to allow for easy comparison between regions. However, the additional plots you have suggested may also be beneficial with aiding the readers understanding. Therefore, we decided to add these additional plots to the supplementary materials. Please see Figures S8, S9 and S10.

Three new figures, S8, S9 and S10 have been added to the revised supplementary materials.